# Comprehension of the Route for the Synthesis of Co/Fe LDHs via the Method of Coprecipitation with Varying pH

**DOI:** 10.3390/nano12091570

**Published:** 2022-05-06

**Authors:** Chérif Morcos, Alain Seron, Nicolas Maubec, Ioannis Ignatiadis, Stéphanie Betelu

**Affiliations:** 1BRGM, French Geological Survey, 3 Avenue Claude Guillemin, CEDEX 02, 45060 Orleans, France; c.morcos@brgm.fr (C.M.); n.maubec@brgm.fr (N.M.); i.ignatiadis@brgm.fr (I.I.); 2LGC, Chemical Engineering Laboratory, University of Toulouse III, 118 Route de Narbonne, CEDEX 09, 31062 Toulouse, France

**Keywords:** Co/Fe LDH formation mechanism, reaction intermediates, ferrihydrite, Co^II^ sorption, Co/Fe redox reactions

## Abstract

Co/Fe-based layered double hydroxides (LDHs) are among the most promising materials for electrochemical applications, particularly in the development of energy storage devices, such as electrochemical capacitors. They have also been demonstrated to function as energy conversion catalysts in photoelectrochemical applications for CO_2_ conversion into valuable chemicals. Understanding the formation mechanisms of such compounds is therefore of prime interest for further controlling the chemical composition, structure, morphology, and/or reactivity of synthesized materials. In this study, a combination of X-ray diffraction, vibrational and absorption spectroscopies, as well as physical and chemical analyses were used to provide deep insight into the coprecipitation formation mechanisms of Co/Fe-based LDHs under high supersaturation conditions. This procedure consists of adding an alkaline aqueous solution (2.80 M NaOH and 0.78 M Na_2_CO_3_) into a cationic solution (0.15 M Co^II^ and 0.05 M Fe^III^) and varying the pH until the desired pH value is reached. Beginning at pH 2, pH increases induce precipitation of Fe^III^ as ferrihydrite, which is the pristine reactional intermediate. From pH > 2, Co^II^ sorption on ferrihydrite promotes a redox reaction between Fe^III^ of ferrihydrite and the sorbed Co^II^. The crystallinity of the poorly crystalized ferrihydrite progressively decreases with increasing pH. The combination of such a phenomenon with the hydrolysis of both the sorbed Co^III^ and free Co^II^ generates pristine hydroxylated Fe^II^/Co^III^ LDHs at pH 7. Above pH 7, free Co^II^ hydrolysis proceeds, which is responsible for the local dissolution of pristine LDHs and their reprecipitation and then 3D organization into Co^II^_4_Fe^II^_2_Co^III^_2_ LDHs. The progressive incorporation of Co^II^ into the LDH structure is accountable for two phenomena: decreased coulombic attraction between the positive surface-charge sites and the interlayer anions and, concomitantly, the relative redox potential evolution of the redox species, such as when Fe^II^ is re-oxidized to Fe^III^, while Co^III^ is re-reduced to Co^II^, returning to a Co^II^_6_Fe^III^_2_ LDH. The nature of the interlamellar species (OH^−^, HCO_3_^−^, CO_3_^2−^ and NO_3_^−^) depends on their mobility and the speciation of anions in response to changing pH.

## 1. Introduction

Layered double hydroxides (LDHs) are a group of minerals consisting of stacked brucite-like layers. They can be represented by the general formula [M_1−x_^II^M_x_^III^(OH)_2_] (A^n−^)_x/n_, mH_2_O, where M^II^ (e.g., Mg^2+^,Co^2+^, …) and M^III^ (e.g., Fe^3+^, Al^3+^, …) are the divalent and trivalent cations, respectively; *x* is the molar ratio M^III^/(M^II^ + M^III^) [1,2,3,4,5]. According to the abovementioned formula, *m* is the number of water molecules intercalated in the interlamellar space of the LDH structure. The positive charge, due to the presence of the trivalent cations, is balanced by exchangeable anions A^n−^ that are located in the interlamellar space and whose width depends on, among other things, the size and nature of the intercalated anions (CO_3_^2−^, Cl^−^, NO_3_^−^ …). Several synthesis routes, such as coprecipitation, the urea hydrolysis method and the sol–gel method [5,6], are described in the scientific literature for synthesizing LDHs of a broad range of formulations [7] and particle sizes (1 nm < d < 10 µm) with various morphologies [8], leading to a large spectrum of specific surface area values up to ~100 m^2^·g^−1^ [9,10,11]. The synthesis route is also responsible for a diversity of intercalated anions (CO_3_^2−^, Cl^−^, HCO_3_^−^, NO_3_^−^ …) with tunable affinity depending on the anions in dynamic equilibrium with the media, even if CO_3_^2−^ tends to have one of the highest affinities for most LDHs [12]. LDHs are well-known for their anionic exchange capacity (AEC) in interlamellar space [13,14,15], which varies between 2 and 5 meq·g^−1^ [16,17]. For these reasons, LDHs have attracted significant interest for industrial and environmental applications [7,18,19,20,21,22,23]. Moreover, LDHs were recently demonstrated as being among the most promising materials for the photoelectrochemical reduction of CO_2_, due to their capacity to facilitate the elaboration of sophisticated catalysts, among other things [24]. Coprecipitation is the most common synthesis route [4,25,26]. Two methods of coprecipitation are used, involving (i) high supersaturation and (ii) low supersaturation [27]. The low supersaturation method consists of adding a solution containing a mixture of M^II^ and M^III^ cations at the desired molar ratio as precursor salts to an alkaline aqueous solution containing the desired anions and maintaining the pH at the selected value. The high supersaturation method consists of adding both a titrant into the cationic solution and the alkaline aqueous solution containing the anion to be intercalated, and the pH is varied until the desired value is reached. This last method promotes nucleation at the expense of particle growth and consequently produces crystallites of small sizes [28,29].

Many studies have been conducted in order to understand the coprecipitation processes and mechanisms leading to LDH formation. For the synthesis of LDHs containing Mg/Al, Mg/Fe, Co/Al and Cr/Zn, Boclair et al. [30,31] found that two buffered pH ranges (plateaus) are observed during the titration of the solution containing both divalent and trivalent cations using an alkaline solution. Hydroxyl anions cause the trivalent cation to precipitate as low-soluble hydroxide or oxyhydroxide during the first plateau. During the second plateau, hydroxyl anions are responsible for the coprecipitation of M^II^ and M^III^ cations, resulting in LDH formation. The pH value range of this second precipitation event is located between those of M^II^ and M^III^ precipitation.

For Mg/Al and Ni/Fe LDHs, Radha et al. [32] proposed a mechanism consisting of the precipitation of a trivalent cation into an amorphous phase that dissolves before the coprecipitation of divalent and trivalent cations for the obtainment of LDHs. For Ni/Fe LDHs, Grégoire et al. [33] identified the precipitation of akaganeite (β-Fe^III^O(OH,Cl)), occurring at the beginning of the titration, as a precursor for the first pH plateau. The presence of Ni^2+^ in the medium prevents the transformation of akaganeite to goethite. Indeed, the progressive pH increase leads to akaganeite being negatively charged. In such conditions, [Ni(H_2_O)_6_]^2+^ reacts with the hydroxyl surface of the akaganeite to form a cationic complex [Ni(OH)(H_2_O)_5_]^+^. The hydroxylation of this cationic complex continues until the appearance of a neutral complex [Ni(OH)_2_(H_2_O)_4_], which forms a lamellar α-[Ni(OH)_1−__δ_(H_2_O)_δ_]^+^ phase via an olation reaction [34]. This positive phase may be adsorbed on the oxygen atoms of the hydroxyl groups of the negative ferric phase. This adsorption may assist the diffusion of iron from the akaganeite to the Ni hydroxide phase, thus generating the LDH layers. After the formation of the Ni/Fe LDHs, two hydroxide phases, β-Fe^III^O(OH,Cl) and α-[Ni(OH)_1−__δ_(H_2_O)_δ_]^+^, were reported [13] using Raman spectroscopy, even after a hydrothermal treatment.

Since the 2010s, Co/Fe LDHs have been extensively investigated as efficient, reliable and robust catalysts or supercapacitors [35,36,37,38,39,40,41,42,43]. The presence of the Co^III^/Co^II^ pair in LDHs significantly improves the conductivity and reversibility of electron transfer [44,45,46]. Integrated into the formulation of LDHs, Fe^III^ promotes electronic transfer via the hopping mechanism, due to its electron-acceptor nature [47,48]. The use of Co-based LDHs as electrophotocatalyzers was also demonstrated in the conversion of atmospheric CO_2_ into CH_4_ [49]. Consequently, compounds based on cobalt and iron represent a major stack in the development of advanced technologies. Understanding the formation mechanisms of such compounds is thus of prime interest.

To the best of our knowledge, this study is the first to investigate the formation mechanisms of Co/Fe LDHs by coprecipitation via the high supersaturation method at 35 °C [7]. Both pH and conductivity were monitored throughout the titration for investigation of the physical and chemical parameters of the solution, while atomic absorption spectroscopy was used to determine the residual concentrations of the M^II^ and M^III^ cations.

The powder X-ray diffractometry (PXRD) characterization of the solids obtained at different pH levels allowed us to determine their structure, crystallinity and purity. The nature of the interlamellar anions was studied by FTIR (Fourier transform infrared) spectroscopy.

## 2. Experimental Section

### 2.1. Chemicals

All chemicals (>99.8% purity) used for the Co/Fe LDH synthesis were supplied by Merck-Sigma-Aldrich (Merck KGaA, Darmstadt, Germany): Co(NO_3_)_2_·6H_2_O, Fe(NO_3_)_3_·9H_2_O, Na_2_CO_3_and NaOH. KBr (>99% purity) used for FTIR measurements was supplied by Merck-Sigma-Aldrich. An Fe standard solution of 1000 ppm (Fe(NO_3_)_3_ in 0.5 M HNO_3_, Certipur^®^, Burlington, MA, US), ammonium acetate (ACS reagent, ≥97%), acetic acid (ACS reagent, ≥99.7%), hydroxylamine hydrochloride (ReagentPlus^®^, Burlington, MA, US, 99%) and 1,10-phenanthroline hydrochlorate (ReagentPlus^®^, 99%) were used for UV–visible spectrophotometric measurements.

### 2.2. Coprecipitation Methodology, Solids Purification and Aging

The coprecipitation of trivalent Fe and divalent Co cations was investigated via the continuous injection of an alkaline titrant into a cationic aqueous mixture in the pH range of 2 to 11 until reaching the desired pH end value. Nine titration experiments, corresponding to nine different pH end values, were performed using 100 mL solutions of Co^II^ (0.15 M) and Fe^III^ (0.05 M). The initial pH value of the Co^II^/Fe^III^ solution was 1.8. The titrant used was a solution of NaOH (2.80 M) and Na_2_CO_3_ (0.78 M).

The titrant was added accurately at 3.9 ± 0.1 µL·s^−1^ using an automatic titrator (Metrohm model Titrando; technology Dosino, Herisau, Switzerland) under vigorous stirring (500 rpm). Titrations were conducted under atmospheric air and pressure in a thermoregulated 150 mL cylindrical Pyrex double-walled water-jacketed reactor at 35 ± 0.2 °C (DC10-P5/U thermostat bath, Haake, Vreden, Germany). During titrations, pH values (±0.3%) of the titrated solutions were recorded as a function of the injected volume of titrant solution; measurements of conductivity and temperature were also continuously recorded with a data-acquisition system (Keithley Instruments, model 2700, Cleveland, OH, USA) controlled by a computer equipped with KickStart 2.5.0 software from Keithley Instruments.

After the pH reached the desired end value, the obtained slurries were centrifuged at 4000 rpm for 15 min to eliminate the supernatants containing the remaining reagents. Then, the resulting slurry underwent 5 dialysis steps of 24 h each in milliQ water (18 MΩ·cm at 25 °C) to remove residual reagent. To allow for the structural characterization of the solids, powders were obtained by freeze drying the slurry at −80 °C and 0.02 mbar using a Christ Alpha 2–4 LSCPlus freeze dryer (Martin Christ Gefriertrocknungsanlagen GmbH, Osterode am Harz, Germany).

For optimal use in the intended applications, LDH particles should be stored as slurries to facilitate handling and hydrodynamic transport and avoid single particle agglomeration. Consequently, synthetized solids were stocked as particles dispersed into fresh milliQ water at room temperature in sealed polyethylene flasks. The influence of aging (700 days) was investigated in such specific conditions.

### 2.3. Physical and Chemical Characterization of the Solids

Obtained solids were ground, using an agate pestle and mortar. In order to determine the nature of the crystalline phases, PXRD analyses were recorded from 5 to 75° in 2θ by using a Bruker D8 Advance diffractometer (Palaiseau, France) using a copper anticathode (λ Kα ≈ 0.1542 nm). The lattice parameters were calculated using the following Equations (1)–(3):(1)$$\frac{1}{{d_{hkl}}^{2}}=\frac{h^{2}}{a^{2}}+\frac{k^{2}}{b^{2}}+\frac{l^{2}}{c^{2}}\;$$
for orthorhombic crystalline systems,
(2)$$\frac{1}{{d_{hkl}}^{2}}=\frac{h^{2}+k^{2}}{a^{2}}+\frac{l^{2}}{c^{2}}\;$$
for trigonal crystalline systems, and
(3)$$\frac{1}{\;d_{hkl}^{2}}=\frac{4\left(h^{2}+hk+k^{2}\right)}{{3a}^{2}}+\frac{l^{2}}{c^{2}}\;$$
for hexagonal crystalline systems.

The average LDH crystallite sizes in the direction perpendicular to the lattice planes were calculated using the Debye–Scherrer equation with $t=\frac{K\lambda }{\beta \cos\theta_{hkl}}$, where *t* is the crystallite size (nm), *K* is Scherrer’s number (usually taken as 0.9), *λ* is the incident wavelength (0.1542 nm), *β* is the full width at half maximum (FWHM) value of the peak and *θ_hkl_* is the Bragg angle of the peak. Let us note that the particles are mainly composed of stacked crystallites; thus, sizes observed by scanning electron microscopy (SEM) analyses are expected to be higher than those calculated using the Debye–Scherrer formula.

Surface morphology was studied by SEM with secondary electron imaging, using Tescan Mira 3 equipment (Gaithersburg, MD, USA). Synthesized LDH slurries, obtained at pH 7, 8, 9 and 11, were diluted 100 times in water. Then, 30 µL of the diluted suspension was deposited on sample holders and dried at room temperature. Compared to freeze drying, which is responsible for the aggregation/agglomeration of LDH particles, such a procedure results in a very thin film of preserved particles in a broad range of orientations, which provides high-quality images, allowing for the exploration of basal faces as well as particle thicknesses. This way of preparation resulted in a better characterization of the particle morphology.

Atomic absorption spectroscopy (AAS) analyses by means of a Spectra AA 220 FS VARIAN instrument (Le Plessis-Robinson, France) were used in order to quantify the amount of remaining metal cations in the filtered (0.1 µm) and acidified (pH < 1) supernatant.

FTIR transmission spectroscopy (Bruker Equinox 55 FTIR spectrophotometer) was performed in the range 4000–400 cm^−1^ on pellets prepared using 150 mg of KBr and 0.5 mg of solid powder, which were pressurized using a manual hydraulic press of 10 tons (Grasbey Specac, Orpington, UK).

Ultraviolet–visible (UV–visible) spectrophotometric measurements were performed using a Varian UV/visible Cary 50 spectrophotometer and a cuvette of 10 mm for path optical length. It was used for iron speciation determination. A UV–visible spectrum of [(C_12_H_8_N_2_)_3_Fe]^2+^ complex was recorded in the range from 800 to 300 nm for Fe concentrations ranging from 1 to 5 mg·L^−1^. For this purpose, Fe^III^ standard solutions were diluted in a reducing and a complexing acetate buffer of 0.1 M and pH 5.6. Hydroxylamine was used as an Fe^III^ reducer and 1,10-phenanthroline as a stable Fe^II^ complexing agent. The reducing and complexing buffer consists of 25 mL of the acetate buffer containing 0.2 g·L^−1^ of 1,10-phenanthroline and 14 g·L^−1^ of hydroxylamine. The calibration curve was established by plotting the absorbance measured at λ_Max_ = 510 nm versus the Fe^2+^ concentration. For solid analyses, 10 mg of solids was dissolved into 10 mL of HNO_3_ 4% (36.14 g·L^−1^ ≈ 0.57 mol·L^−1^). Then, 700 µL of the latter solution was added to 25 mL of the complexing buffer with or without 1,10-phenanthroline. Fe^II^ was determined in the absence of hydroxylamine. The total Fe was determined in the presence of 1,10-phenanthroline. Following Beer–Lambert law, the absorbance coefficient ε was deduced from the slope of the plotted absorbance calibration curve at λ_Max_ = 510 nm as a function of the molar concentration of [(C_12_H_8_N_2_)_3_Fe]^2+^ and is equal to 12,048 mol·L^−1^·cm^−1^ with R^2^ = 0.9995.

### 2.4. Geochemical Modeling

The PHREEQC^®^ (Orleans, France) geochemical code was used for thermodynamic investigation based on the associated THERMODEM^®^ (Orleans, France) thermodynamic database generated by BRGM. The geochemical model predicted the evolution of the species concentration during the modeled titration of dissolved Co(NO_3_)_2_ (0.15 M) and Fe(NO_3_)_3_ (0.05 M) by the alkaline titrant solution of NaOH (2.80 M) and Na_2_CO_3_ (0.78 M) at 35 °C, according to equilibria with ferrihydrite 2 line, ferrihydrite 6 line and Co(OH)_2_ (Appendix A). The conductivity values were calculated according to the following formula:Conductivity (at T °C) = ([C] × Λ°m) × (1 + 0.02 × (T − 25))
where [C] is the concentration of the involved species in mol.m^−3^, Λ°m is the value of the ionic conductivity at 25 °C and at infinite dilution (Ω^−1^·m^2^·mol^−1^) and T is the temperature in °C.

The shape of the modeled conductivity is not in agreement with data measured during the experimental titration. Indeed, the conducted modeling does not take into account the thermodynamic data related to adsorption of Co^2+^ on ferrihydrite and Na^+^ on precipitates nor coprecipitation of LDHs, because these data are absent in the database used. However, the intercomparison between experimental and modeled titrations enabled interpretation of the different phenomena that take place during the experimental titration.

## 3. Results and Discussion

### 3.1. Titration

Figure 1 shows an example of the titration curve of the mixed Co${\left({\mathrm{NO}}_{3}\right)}_{2\left(\mathrm{aq}\right)}$ (0.15 M) − Fe${\left({\mathrm{NO}}_{3}\right)}_{3\left(\mathrm{aq}\right)}$ (0.05 M) solution by the alkaline titrant solution from the initial pH to 11.

The initial media solution is acidic (pH~1.8) due to the constituent cations, which behave as weak Bronsted acids and consequently have the ability to produce hydronium (H_3_O^+^ or H^+^_(aq)_) ions. This acidity is a function of the size, charge and electronegativity of the cations. The smaller the cation and the higher the electronegativity of the metal ions, the more acidic it will be [50].

When adding the first drops of the alkaline titrant into the solution of the metallic cations, pH increases slowly, in agreement with H^+^ consumption by both OH^−^ and CO_3_^2−^ (for the formation of H_2_O and H_2_CO_3_, respectively). Fe^3+^ hydrolysis begins (into Fe(OH)^2+^ and Fe(OH)_2_^+^) at the same time. At pH~2.1, pH reaches a first pseudo-plateau in agreement with the simultaneous precipitation of Fe^III^. At around pH~2.3, the precipitation of Fe^3+^ is mostly complete. This first pH plateau is followed by a rapid increase in the pH until 4.1–4.3, i.e., rapid consumption of H^+^ mainly by free OH^−^ and CO_3_^2−^. Beginning from this pH range, HCO_3_^−^ appears, and the equivalent point of the H_2_CO_3_/HCO_3_^−^ acid–base couple is reached at 6.4. From this point, another pseudo-buffered area starts in agreement with Co hydroxylation, which is responsible for the progressive coprecipitation of the metallic cations. The progressive enrichment in Co into the coprecipitated material induces positive charges, which are compensated for by the intercalation of hydrated anions, resulting in LDH formation [30,31]. At around pH~9.0, Co/Fe LDH co-precipitation is complete. The equivalent points of the HCO_3_^−^/CO_3_^2−^ acid–base couple are reached at the pH value of 11. Finally, while the alkaline titrant solution addition is carried out, pH increases up to the pH value of the alkaline solution itself (pH > 11).

According to the ionic composition of the metallic solution, at pH 1.7 and at 35 °C, the conductivity should be equal to 69 mS·cm^−1^, in which NO_3_^−^ accounts for 38.6 mS·cm^−1^, H^+^ for 8.5, Fe^3+^ for 12 and Co^2+^ 9.90 mS·cm^−1^.

The initial measured conductivity was 38 mS·cm^−1^ because the concentration of the dissolved salts exceeds 10^−2^ M, which is the highest limit of applicability for Kohlrausch’s law. Nevertheless, conductivity was used as an indicator to investigate the evolution of the physical and chemical parameters during titration. The recorded conductivity is plotted in Figure 2 (orange line) versus the added volume of alkaline titrant solution.

For better insight, Figure 2 highlights the conductivity variations versus pH values, showing that conductivity greatly fluctuates in the pH ranges 1.8–2.6 during Fe^III^ precipitation (Appendix A) and 6.6–7.8 during LDH coprecipitation. Such unpredictable behavior clearly shows the effects of the concomitant phenomena-inducing transfer of ionic species into solution, in agreement with rapid and instable precipitation–dissolution phenomena. During titration, conductivity is dominated by the presence of NO_3_^−^_,_ whose concentration (0.45 M) does not significantly evolve over time (Appendix A).

As the conductivity linked to the concentration of NO_3_^−^ anions is constant over time, the observed conductivity decrease in the range from 1.8 to 2.6 should predominantly be related to Fe^3+^ precipitation and, to a lesser extent, to H^+^ neutralization by the added OH^−^ and Na^+^, whose concentrations continuously increase (Appendix A). In the pH range from 2.3 to 3.3, the H^+^ neutralization that proceeds should be the only phenomenon weakly influencing conductivity. According to the decrease in cobalt concentration from pH 2 to 9 (Figure 2A), it is apparent, from Figure 2B, that Co^2+^ concentration is responsible for the evolution of the conductivity value. From pH > 2 (Figure 1, orange line), in comparison with Appendix A, any significant decrease in conductivity is thus related to Co^2+^ behavior. The increase in conductivity should be related to the gradual increase in Na^+^ concentration and the increase in total inorganic carbon and, to a lesser extent, OH^−^ ions in the solution (Appendix A). Surprisingly, conductivity only increases from pH 8, whereas it was expected to have increased from pH 5 following the precipitation of Fe^3+^ and the neutralization of H^+^, where Na^+^ concentration reached 0.2 M (10 mS·cm^−1^). At pH 8, the Na^+^ concentration reached 0.6 M (34 mS·cm^−1^). The first decrease in conductivity is in the pH range from 2.3 to 6, and then the plateau observed from pH range 6 to 8, demonstrating that Co^2+^ is accountable for the conductivity evolution from pH 2.3 to 8. The absence of any increase in conductivity in this pH range tends to demonstrate the Na^+^ sorption phenomenon. In agreement with the literature [51], the ionic radii of cations, related to that of oxygen, determines the possibility of Na^+^ insertion into an octahedral lattice without further critical distortion. A cation/O^−II^ radii ratio between 0.414 and 0.732 would indicate the possible insertion of the cation [51,52,53]. The ionic radius of Na^+^ is 0.95 Å, and that of O^−II^ in an octahedral site is 1.40 Å. Na^+^ can thus be inserted into an octahedral lattice (r(Na^+^/O^II^) of 0.68 Å). In addition, because of the high concentration of Na^+^, it should also adsorb onto the external surfaces of Fe^III^ precipitates, in agreement with the pH values being higher than the point of zero net charge (PZNC) [54]; indeed, above PZNC, the solid surface charges are negative. In addition, in the pH range of 6–8, the increase in HCO_3_^−^ in solution should slightly increase conductivity up to ~4 mS·cm^−1^ (Appendix A). The stability of conductivity in such a pH range is linked to HCO_3_^−^ intercalation into the interlamellar space of LDHs. Above pH 8, the increase in conductivity is linked to the increase in both Na^+^ and CO_3_^2−^ concentrations due to the addition of alkaline solution.

### 3.2. Atomic Absorption Spectroscopy Analyses of Co^2+^ and Fe^3+^ in the Supernatant

For each titration pH value, the supernatant extracted from the centrifugation was analyzed by AAS to quantify the remaining amount of Co^2+^ and Fe^3+^ (Figure 2A).

A small decrease in the Fe^III^ concentration at pH 2 indicates the point at which Fe^III^ begins to precipitate. From pH 2 to 3, the sharp decrease in the Fe^III^ solution concentration is in agreement with the precipitation of iron species. Surprisingly, the Fe^III^ concentration remains quasi-constant from pH 3 to 4 at 2.7 × 10^−3^ M, which is close to 5% of the initial Fe^III^ concentration. This concentration value decreases progressively and is close to zero at pH 8. According to the literature, K_s_ values of expected precipitated iron species are usually extremely low (10^−44^–10^−37^) [55,56]. The AAS results suggest the presence of free Fe^2+^ in the media.

Regarding Co^II^, its concentration remains constant until pH 2 (0.15 M). From pH 2 to 3, the Co^2+^ concentration sharply decreases, with a loss of 15% from the initial concentration (from 0.15 to 0.127 M). This result suggests Co^2+^ adsorption onto iron precipitates; the Co/Fe ratio in the solid, i.e., the molar ratio of adsorbed Co^II^ onto precipitated Fe^III^, reaches 1/2. From pH 3 to 5, the Co^II^ concentration remains constant. Beyond this pH value, the Co^II^ concentration decreases progressively; the speed of this Co^II^ concentration decrease accelerates from pH 7 to 8. In the pH range between 2.3 and 8.0, the decreasing evolution of the remaining Co^II^ in the solution is synchronous with the conductivity values. The Co^II^ concentration is close to 0 from pH 9. Measurements of Co^2+^ concentration by AAS and of conductivity enable the calculation of Co^2+^ concentration over time (Appendix A).

### 3.3. PXRD Characterization of Solids Synthesized by Acid—Base Titration

Solids obtained after each of the nine partial titrations were characterized by PXRD to determine the nature of the obtained crystalline phases (Figure 3); when the LDHs were obtained, their lattice parameters were calculated using the Rietveld refinement method (Table 1).

As revealed by PXRD, goethite is the main mineral phase obtained at pH 2 (JCPDS 01-073-0522 (I)), in agreement with diffraction peaks located at 17.85°, 21.1°, 33.16°, 34.62° and 36.58°, which enable us to calculate *d_hkl_*. From the collected XRD data and the orthorhombic crystalline system, the calculated lattice parameters (*a* = 4.38 Å, *b* = 9.93 Å, *c* = 3.10 Å) are in agreement with the literature [57]. The goethite diffraction bands are superimposed on two wide diffraction bumps in the range from 25° to 45° and in the range from 55° to 70°, showing the presence of a poorly crystalline ferrihydrite phase. Indeed, ferrihydrite is a form of iron oxyhydroxide, which usually forms in the course of iron mineralization with relatively poor crystallinity [58]. Ferrihydrite is thermodynamically unstable and can readily recrystallize to goethite, magnetite and lepidocrocite [58,59,60]. The presence of goethite is in agreement with the thermodynamic transformation of ferrihydrite to goethite and is probably linked to washing and/or drying steps.

From pH 3 up to 6, poorly crystalline ferrihydrite is the only phase evidenced by PXRD diffraction peaks located at 35.42°, 39.87°, 45.61°, 52.81° and 62.09°, in agreement with (JCPDS 01-073-8408). The obtainment of ferrihydrite in such a range of pH values clearly demonstrates the artefact observed at pH 2, which is related to the transformation of metastable ferrihydrite to goethite. Ferrihydrite is thus the pristine reactional intermediate.

According to Maillot et al. [61], ferrihydrites synthesized in this study are 5 line structures, at pH 3 and 4; a 4 L structure was observed at pH 6. This change in crystallinity between the so-called 2 line and 6 line ferrihydrite is generally interpreted as a change in mean particle size from 2 to 6 nm, respectively [62,63,64].

The higher the pH, the lower the relative intensity of diffraction bands, i.e., the lower the ferrihydrite crystallinity. From pH 3 to 6, ferrihydrite particle size is calculated from the peak at 35°. It is 9.3, 7.4 and 3.2 nm at pH 3, 4 and 6, respectively. The higher the pH, the smaller the particle size.

Ferrihydrite features small particle sizes and a large specific area. It is, moreover, modeled as a mixture of a defect-free phase and a defective phase, which ranges from 33 to 50% in a 6L ferrihydrite [65]. Ferrihydrite is thus well-known and used for its high surface reactivity, such as in metal adsorption [66,67,68], probably due to the presence of the defective phase. Moreover, the ability to bind metal was demonstrated to slow down [69] and impede [70] the rate of ferrihydrite transformation into other crystalline iron phases. Such behavior is also explained by metal substitution [71,72] in the ferrihydrite structure during dissolution–reprecipitation phenomena. In comparison with the evolution of ferrihydrite into goethite observed at pH 2, and in agreement with the results provided by AAS and conductivity in the pH range 3–6, the maintenance of the ferrihydrite structure is attributed to the presence of both Na^+^ and Co^2+^ in the medium, which can adsorb onto ferrihydrite. As previously discussed, Na^+^ can insert into octahedral lattices and/or bind onto the negatively charged =M–O^−^ surface sites of ferrihydrite. Co^2+^ (ionic radius 0.072 Å) may be able to substitute Fe^3+^ (ionic radius 0.067 Å) in agreement with dissolution–reprecipitation phenomena. Co^2+^ adsorption onto negatively charged =M–O^−^ surface sites is, nevertheless, the most kinetically probable hypothesis. Both Co^2+^ and Na^+^ compete for such adsorption sites. The cation-exchange process favors the uptake of divalent over monovalent species. Co^2+^ is thus the predominant sorbed cation until Na^+^ reaches a concentration value high enough to compete with Co^2+^. It is thus assumed that both Co^2+^ and Na^+^ sorption on the negatively charged =M–O^−^ surface sites of ferrihydrite nuclei generate passivation layers, which significantly slow down the degradation of ferrihydrite to goethite by limiting the iron transfer between sites.

From pH 7, PXRD patterns show characteristic Co/Fe LDH structures [73,74,75] associated with some impurities. The reflection peaks at 17.5°, 20.0°, 24.2°, 26.7°, 33.8°, 35.4° and 39.5° are characteristic of Co(CO_3_)_0.5_(OH)·0.11 H_2_O (JCPDS 00-048-0083) and those at 18.4°, 30.2°, 35.6°, 37.2°, 43.2°, 53.6° and 57.2° are characteristic of the Co/Fe-based spinel structure in the solid (JCPDS 01-083-6167). The position of the LDH diffraction peaks, indexed in Table 1, are in agreement with a hexagonal structure (JCPDS 04-017-8816).

The Rietveld refinement method applied to the XRD pattern (Appendix A) indicates that the materials synthesized at pH 7, 8, 9 and 11 are composed of a 73% LDHs and 27% spinel phase; 54% LDHs, 41% cobalt hydroxycarbonate hydrate and 5% spinel phase; 53% LDHs, 43% cobalt hydroxycarbonate hydrate and 4% spinel phase; 70% of LDHs, 24% cobalt hydroxycarbonate hydrate and 6% spinel phase. In the range from pH 8 to 9, the passivation phenomenon induced by Na^+^ sorption onto the negatively charged =M–O^−^ surface sites of LDH nuclei could be responsible for the precipitation of cobalt hydroxycarbonate hydrate impurities. This adsorption phenomenon is supported by the characteristic constancy of the conductivity values in the pH range 7–9.5. From pH 9 to pH 11, Ostwald maturation through the dissolution–reprecipitation phenomena, which progresses with time and with the addition of alkaline solution [28,76,77], is probably responsible for the increase in both LDH content and crystallinity.

The (110) peaks at ~59° allowed for the determination of the *a* lattice parameter value, which is equal to double the distance at (110); *a =* 2 *× d*(110). This value is directly related to the distance between adjacent cations in the LDH structure. The variation in synthesis pH did not have any effect on the value of the ‘a’ lattice parameter, which was constant at 3.10 ± 0.10 Å from pH 7 to 11.

The (003) peak at ~11° allowed for the determination of the basal spacing (*c*′), as *d*_003_ is directly equal to the thickness of one brucite-like layer and interlayer spacing. As shown in Table 1, d_003_ remains constant with the increase in the pH synthesis value; the interlayer space *c′* ranges between 7.63 and 7.69 Å. According to the literature, the basal spacings correspond to a predominant CO_3_^2−^ interlayer anionic species [18].

Moreover, the positions of the reflections (012)–(015) at intermediate angles (35°–45°) help in calculating the c parameter using the equation linking the Miller indices to the lattice parameter in a hexagonal crystalline system [74]. The ratio between the *c* parameter and the *c*′ reflects the layer periodicity along the c axis. For the solids obtained at pH 7, 8, 9 and 11, the ratios are 2.91, 2.92, 2.91 and 2.94, respectively. In agreement with the literature [1], convergence of the *c*/*c*′ ratio to 3 reflects the perfect periodicity along the *c* axis at a structural scale and confirms the homogeneous distribution of interlayer anion species.

Regardless of the pH, two groups of crystalline LDH particles were identified: a group of pseudo-crystalized particles in which crystallite sizes range from 1.8 to 5.3 nm, and a group of crystalized particles in which crystallite sizes range from 6.0 to 12.2 nm. Their rather different apparent sizes, i.e., 4.46, 3.98, 2.54 and 8.25 nm for pH 7, 8, 9 and 11, respectively, influence the apparent crystallinity of the solid. The decrease in the apparent crystallite sizes from pH 7 to 9 is in agreement with the incorporation of residual Co^II^ into the pristine LDHs precipitated at pH 7. From pH 9 to pH 11, the increase in the apparent crystallite size is in agreement with the Ostwald maturation phenomenon through the dissolution–reprecipitation phenomena, which progresses with time and the addition of the alkaline solution [28,76,77].

### 3.4. LDH Characterization after Aging at Room Temperature

Prior to any characterization, the pH values of the slurries were measured; the exact pH values were 6.98, 7.96, 8.97 and 10.95. The slurries were buffered by the synthesized LDHs.

#### 3.4.1. PXRD of the Solids Collected after Aging at Room Temperature

Figure 4 shows the LDH PXRD patterns after aging at room temperature for 700 days, and Table 2 presents the lattice parameters calculated using the Rietveld refinement method (Appendix A).

As previously observed, PXRD patterns show characteristic Co/Fe LDH structures [73,74,75]. The persistence of cobalt hydroxycarbonate hydrate was demonstrated for materials synthesized at pH 8 and 9. At pH 11, the reflection peaks at 20.6°, 37.0°, 48.9 and 50.8° are characteristic of CoO(OH) (JCPDS 01-072-2280), and those at 18.5°, 30.4°, 35.8°, 37.5°, 43.5°, 54.0° and 57.5° are characteristic of a cobalt-based spinel structure in the solid (JCPDS 01-083-6167).

The Rietveld refinement method applied to the XRD pattern (Appendix A) indicated that the material synthesized at pH 7 is composed of 100% LDHs. At pH 8 and 9, 69% and 58% of the synthesized materials are LDH materials, respectively, and the rest are composed of cobalt hydroxycarbonate hydrate. At pH 11, the synthesized material is composed of 75% Co/Fe LDH, 21% cobalt iron oxyhydroxides and 4% cobalt-based spinel phase.

The LDH basal spacings, ranging from 7.60 to 7.71 Å, are correlated with the predominance of CO_3_^2−^ as anionic species in the interlayer space [18]; the d_003_ value of 8.71 Å obtained at pH 7 is in agreement with the predominance of NO_3_^−^ species [1].

Regardless of the investigated pH, aging favors crystallinity. Particle sizes are drastically increased at pH 8 and 9, which tends to demonstrate that the higher the defect contents, the higher the degree of Ostwald maturation, which also favors particle growth.

#### 3.4.2. Scanning Electron Microscopy (SEM) of Co/Fe LDHs Synthesized at pH 7, 8, 9 and 11

Figure 5 shows SEM secondary electron imaging of the Co/Fe LDHs synthesized at pH 7, 8, 9 and 11.

The Co/Fe compounds resulting from synthesis at pH values 7, 8, 9 and 11 are composed of agglomerated platelet particles presenting a pseudo-hexagonal shape. The hexagonal morphology of these particles is in good agreement with the usual morphology of LDH particles [78,79,80]. The particles show hexagon side lengths from 100 to 500 nm and thicknesses between 10 and 50 nm (Table 3).

#### 3.4.3. FTIR Spectra of the LDHs Synthesized at pH 7, 8, 9 and 11

Figure 6 shows the FTIR spectrum of the LDHs synthesized at pH 7, 8, 9 and 11 in the range from 400 to 800 cm^−1^.

Two characteristic peaks are associated with the presence of Co/Fe LDH in the solid: 490 cm^−1^ for Fe–O and 515 cm^−1^ for Co–OH vibrations [81,82]. The higher the crystallinity, the better the band resolution. The band at 575 cm^−1^ is characteristic of Co–O vibration in Co–O(OH) [83]. The presence of the Co-based spinel is manifested by the characteristic Co–O vibration bands at 553 and 655 cm^−1^ [82,84,85].

Figure 7 shows the FTIR spectrum of the LDHs synthesized at pH 7, 8, 9 and 11 in the range from 1800 to 1100 cm^−1^.

The bands observed at 1640, 1540, 1380 and 1352 cm^−1^ are related to the presence of interlamellar water [86], hydrogenocarbonates (HCO_3_^−^) [12,30], nitrates (NO_3_^−^) and carbonates (CO_3_^2−^) [12,87,88,89,90], respectively. Surprisingly, NO_3_^−^ and CO_3_^2−^ are the only interlamellar anions observed at pH 7. As it is quite difficult to differentiate the hydroxyl anions in FTIR, OH^−^ could be the predominant interlamellar anion. From pH 8 to 11, the FTIR spectra show a drastic increase in both CO_3_^2−^/NO_3_^−^ and CO_3_^2−^/HCO_3_^−^ band ratios, while the NO_3_^−^/HCO_3_^−^ band ratio decreases. Data provided by FTIR support that HCO_3_^−^ is predominant at pH 8 and 9, while CO_3_^2−^ is the predominant interlamellar anion at pH 11.

#### 3.4.4. UV–Visible Absorbance of Various Contents of Fe and Synthesized LDHs

Figure 8 shows the plotted UV–visible spectra of 1, 2, 3, 4 and 5 mg·L^−1^ standard solutions of [(C_12_H_8_N_2_)_3_Fe]^2+^ complex (dotted lines). It also shows the UV spectra of the dissolved Co/Fe LDHs synthesized at pH values 7, 8 and 9 (solid lines). The measurements were conducted in the presence or the absence (mentioned “without R”) of the Fe^III^ reducing agent from 325 to 625 nm.

The [(C_12_H_8_N_2_)_3_Fe]^2+^ complex spectra present four absorption bands at 511, 476, 440 and 408 nm [91], which are characteristic of a low spin complexed Fe^II^. The absorbance recorded for the dissolved LDHs shows the same shape as the analyzed [(C_12_H_8_N_2_)_3_Fe]^2+^ complex, regardless of the analyzed sample and whether Fe^III^ reducing agent is present or absent. These results tend to demonstrate that all the LDH-iron is present as Fe^II^. Nevertheless, the addition of Co^II^ to a [(C_12_H_8_N_2_)_3_Fe]^3+^ complex standard solution, in which (C_12_H_8_N_2_)_3_ complexes Fe^III^ at concentrations equimolar to Fe^III^, is responsible for the progressive total reduction of [(C_12_H_8_N_2_)_3_Fe]^3+^ to [(C_12_H_8_N_2_)_3_Fe]^2+^, which strongly colors the solution.

In any case, the Fe^II^ concentration in the solutions is two times greater when prepared from LDHs synthesized at pH 7 than from LDHs synthesized at pH 8 and 9. These results are in good agreement with AAS measurements conducted on the LDH synthesis supernatants and support the identity of the LDHs synthesized at pH 7 as being M^II^/M^III^ 2/2 and those at pH 8 and 9 as being 3/1.

## 4. Discussion

The monitoring of pH and conductivity carried out during titrations completed by the PXRD characterizations of solids, UV–visible characterization of dissolved solids and AAS analyses on the supernatants allowed for the determination of the speciation of intermediates involved in the pathway to Co/Fe LDH formation.

From pH 1.8 to 2.6, there is precipitation of the trivalent cation Fe^III^. In the pH range 1.8–6, the formation of ferrihydrite was confirmed. Co^2+^ and Na^+^ sorption on the negatively charged =M–O^−^ surface sites of ferrihydrite nuclei generate passivation layers that maintain the ferrihydrite structure by avoiding degradation to goethite via limiting of iron transfer between nuclei.

From the AAS measurements, the adsorption capacitance of the Co^2+^ on the ferrihydrite at pH 6 was evaluated using Equation (4):Q = (C_0_ − C_t_) × V/m (4)
where C_0_ and C_t_ are the initial and equilibrium mass concentrations of cobalt, V is the volume of the solution and m is the weight of the ferrihydrite deduced from precipitated iron based on the chemical formula Fe(OH)_3_ [92]. Q is the adsorption capacity, expressed in mg of sorbed Co^2+^ per g of ferrihydrite, and is equal to 220 mg·g^−1^ at pH 6. This result is in good agreement with the literature [66,68,93,94]. Figure 9 shows the evolution of Co^2+^ concentration versus time in the pH range from 2 to 2.2. Data were extracted from Appendix A. The data enabled the determination of the kinetic constant for Co^2+^ adsorption on ferrihydrite.

Co^2+^ concentration decreases linearly versus time with a regression coefficient (R^2^) of 92%.

The adsorption equilibrium kinetics of Co^2+^ on ferrihydrite is of a zero order (α = 0); this behavior can be explained by the cobalt concentration in the solution, which is three times higher than that of Fe^3+^. The adsorption kinetic constant is close to 1.57 × 10^−6^ mol·min^−1^.

Co^II^ adsorption on ferrihydrite is responsible for the decrease in the Co^III^/Co^II^ redox potential in response to increasing pH, where Co^II^ becomes a reducing agent for Fe^III^ in ferrihydrite [92,95,96,97,98]. When sorbed on ferrihydrite, Co^II^ is oxidized to sorbed Co^III^ in agreement with the electron transfer to Fe^III^ atoms, which are reduced to Fe^II^ [95,97,98,99,100,101]. Simultaneously, both sorbed Co^III^ and free Co^II^ hydrolysis are responsible for H^+^ generation, which facilitates local dissolution of ferrihydrite. Such phenomena are highlighted by conductivity fluctuations in the pH range 6 to 7.5. These latter numbers are in agreement with the progressive decrease in ferrihydrite crystallinity and particle size with increasing pH. The sequential adsorption–reprecipitation occurring on the ferrihydrite surface sites generate the Fe^II^/Co^III^ LDH nuclei clusters. Those positively charged nuclei attract anions. At pH 7, the association of nuclei and anions generate a pristine 3D organized Fe^II^/Co^III^ layered structure. Above pH 7, Co^II^ hydrolysis occurs. It is responsible for the local dissolution of pristine LDHs, reprecipitation and then the 3D organization of Co^II^_4_Fe^II^_2_Co^III^_2_ LDHs, whose particle size and degree of crystalline organization increase along with the pH.

The progressive incorporation of Co^II^ into the LDH structure can account for two concomitant phenomena. The more Co^II^ is integrated into the LDHs, the greater the relative difference that evolves between the redox potentials, such as the redox couple Co^III^/Co^II^ becoming more oxidant than Fe^III^/Fe^II^. This shift allows the reverse redox reaction, such as Fe^II^ oxidizing to Fe^III^, while Co^III^ reduces to Co^II^ within the LDH structure for the obtainment of Co^II^_6_Fe^III^_2_ LDHs. At the same time, the progressive incorporation of Co^II^ into the LDH structure is responsible for the decrease in the coulombic attraction between the positive surface layers and the interlayer anions and, thus, the progressive hydration of the material, which is, in turn, responsible for the progressive increase in the interlamellar space.

The nature of the interlamellar species (OH^−^, HCO_3_^−^, CO_3_^2−^ and NO_3_^−^) depends on the mobility and the speciation of anions with changing pH. Indeed, at pH 7, the LDH could be hydroxylated in agreement with the OH^−^ mobility 5.3 × 10^−9^ m.s^−1^, which is nearly 3.0 and 4.5 times higher than that of NO_3_^−^ and HCO_3_^−^, respectively. At pH 8 and 9, and at pH 11, HCO_3_^−^ and then CO_3_^2−^ are successively the predominant anions in the interlamellar space, in agreement with inorganic carbon speciation mechanisms.

Following the insights that have been provided by these analyses, a mechanism for the formation of the LDHs Co/Fe is proposed and illustrated as follows (Equations (5) and (6)):[Fe^III^(OH)_3_] ⇄ [Fe^III^(OH)O_2_^2−^] + 2H^+^(5)
[Fe^III^(OH)O_2_^2−^] + Co^II^ ⇄ [Fe^III^(OH)O_2_]–Co^II^(6)

Ferrihydrite loses its protons (Equation (5)), and divalent cobalt is sorbed onto the negative-charged surface (Equation (6)). Then, Fe^III^ partly reduces to Fe^II^, simultaneously with Co^II^ oxidizing to Co^III^ (Equation (7)).
[Fe^III^(OH)O_2_]–Co^II^ ⇄ [Fe^III^_1−x_Fe^II^_x_(OH)O_2_]–(Co^II^_1−x_Co^III^_x_)(7)

Furthermore, x progressively increases until reaching 1, such as when [Fe^III^_1−x_Fe^II^_x_(OH)O_2_]–(Co^II^_1−x_ Co^III^_x_) transforms to [Fe^II^Co^III^(OH)O_2_], which reacts with H_2_O, such as in Equation (8):2 [Fe^II^Co^III^(OH)O_2_] + (4 + ε)H_2_O ⇄ [Fe^II^_2_Co^III^_2_(OH)_8_](OH)_2_, εH_2_O, (8)

The obtainment of an Fe^II^/Co^III^ LDH is in agreement with the AAS and UV analyses, with OH^−^ anion predominant in the interlamellar space.

At the same time, Co^II^ in solution hydrolyzes as follows (Equation (9)):[Co^II^(H_2_O)_6_]^2+^ ⇄ [Co^II^(OH)(H_2_O)_5_]^+^ + H^+^
(9)

Such a phenomenon is responsible for local dissolution–reprecipitation and 3D organization, such as Co^II^ progressively integrating into the structure of the pristine LDHs, in agreement with Equations (10) and (11):[Fe^II^_2_Co^III^_2_(OH)_8_](OH)_2_, εH_2_O + 4[Co^II^(OH)(H_2_O)_5_]^+^ + 2OH^−^ + yHCO_3_^−^ + ½ (2 − y)CO_3_^2−^ ⇄ [Co^II^_4_Fe^II^_2_Co^III^_2_(OH)_16_](HCO_3_)_y_(CO_3_)_½(2−y)_,(2ε + ε′)H_2_O + (20 − ε′)H_2_O(10)
[Co^II^_4_Fe^II^_2_Co^III^_2_(OH)_16_](HCO_3_)_y_(CO_3_)_½(2−y)_,(2ε + ε′)H_2_O + xH_2_O ⇄ [Co^II^_6_Fe^III^_2_(OH)_16_](HCO_3_)_y_(CO_3_)_½(2−y)_, XH_2_O(11)
where X = 2ε + ε′ + x, and Fe^II^ oxidizes while Co^III^ is reduced.

With the decrease in y with increasing pH, a solid is formed according to Equation (12):[Co^II^_6_Fe^III^_2_(OH)_16_](HCO_3_)_y_(CO_3_)_½(2−y)_, XH_2_O ⇄ [Co^II^_6_Fe^III^_2_(OH)_16_](CO_3_), XH_2_O(12)

The long-term storage of LDH particles as slurries at room temperature is responsible for the continuation of Ostwald maturation.

## 5. Conclusions

This study focused on the formation mechanisms of Co/Fe LDHs under high supersaturation conditions.

The monitoring of pH and conductivity carried out during titrations, together with PXRD characterization of solids, UV–visible characterization of dissolved solids and AAS analyses of the supernatants, allowed for the determination of the speciation of intermediates involved in the pathway to the formation of the final Co/Fe LDHs.

From pH 2, the pH increase induces Fe^III^ precipitation as ferrihydrite, which is the pristine reactional intermediate. The maintenance of such a structure in the pH range 3–6 is explained by Co^II^ and Na^I^ adsorption onto ferrihydrite. Co^II^ adsorption on ferrihydrite is responsible for modifying the relative position of the redox potential of Co^III^/Co^II^ and Fe^III^/Fe^II^ redox couples, such as when compared to pH, and Co^II^ becomes a reducing agent for Fe^III^ in ferrihydrite. Sorbed Co^II^ is thus oxidized to sorbed Co^III^ in agreement with the electron transfer to Fe^III^ atoms in the ferrihydrite structure, which are reduced to Fe^II^. This phenomenon induces the decrease in ferrihydrite crystallinity. At the same time, at 6 ≤ pH ≤ 7, both sorbed Co^III^ and free aqueous Co^II^ hydrolysis drive the amorphization of ferrihydrite, already in progress, into positively charged Fe^II^/Co^III^ brucite-like nuclei, in agreement with dissolution–reprecipitation phenomena. At pH 7, these charged nuclei organize into a 3D Fe^II^/Co^III^ LDH pristine structure, which is mainly hydroxylated. Above pH 7, Co^II^ hydrolysis proceeds and is responsible for the local dissolution of pristine LDHs and their reprecipitation and then 3D organization into Co^II^_4_Fe^II^_2_Co^III^_2_ LDHs. The progressive incorporation of cobalt into the LDH structure is responsible for the strength inversion of the redox potential of the two redox couples, Co^III^/Co^II^ and Fe^III^/Fe^II^, such as when Fe^II^ (as a stronger reducing agent) re-oxidizes to Fe^III^ while Co^III^ (as a stronger oxidizing agent) is re-reduced to Co^II^, for the obtainment of Co^II^_6_Fe^III^_2_ LDH. The long-term storage of LDH particles as slurries at room temperature is responsible for the continuation of Ostwald maturation. At pH 8 and 9, and at pH 11, HCO_3_^−^ and then CO_3_^2−^ are successively the predominant anions in the interlamellar space, in agreement with inorganic carbon speciation mechanisms.

## Figures and Tables

**Figure 1 nanomaterials-12-01570-f001:**
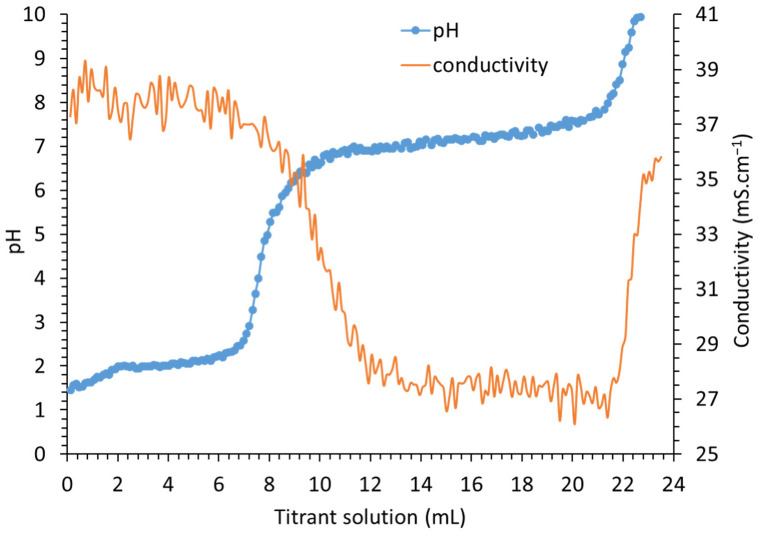
Titration curve of 100 mL of a solution containing the cations mixture Co^2+^ (0.15 M) and Fe^3+^ (0.05 M) using the alkaline titrant solution of NaOH (2.80 M) and Na_2_CO_3_ (0.78 M).

**Figure 2 nanomaterials-12-01570-f002:**
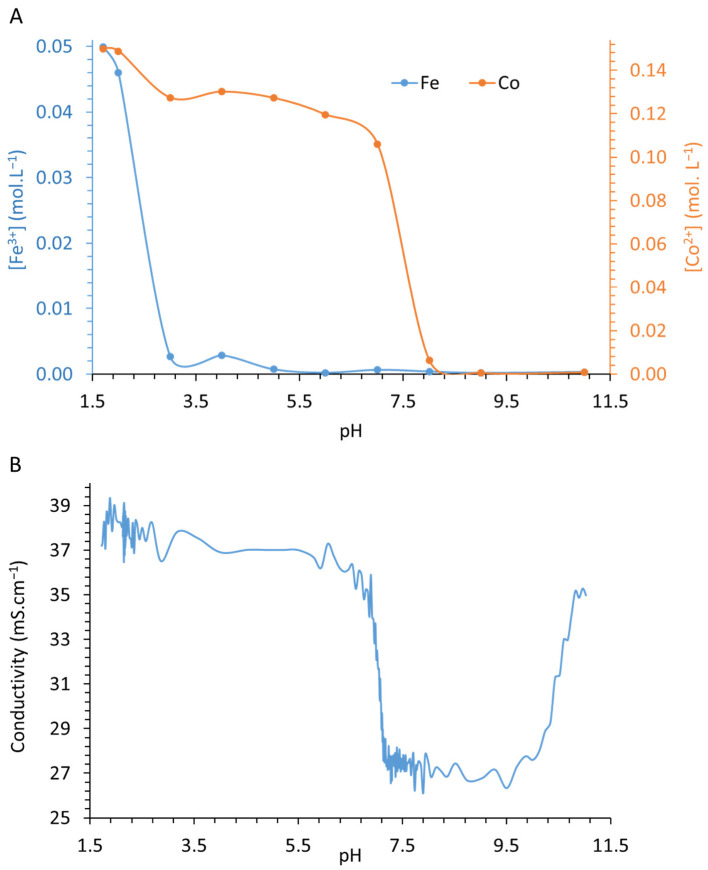
(**A**) Remaining concentrations of Fe (blue) and Co (orange) in the supernatant and (**B**) measured conductivity as a function of the corresponding titration pH.

**Figure 3 nanomaterials-12-01570-f003:**
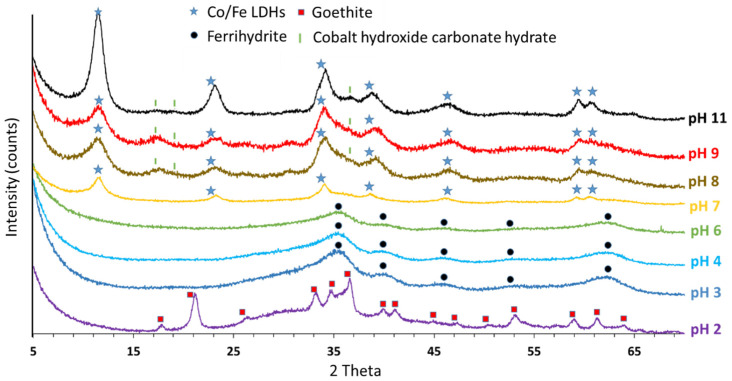
PXRD pattern for solids collected after titrating the starting solution up to the reported pH values.

**Figure 4 nanomaterials-12-01570-f004:**
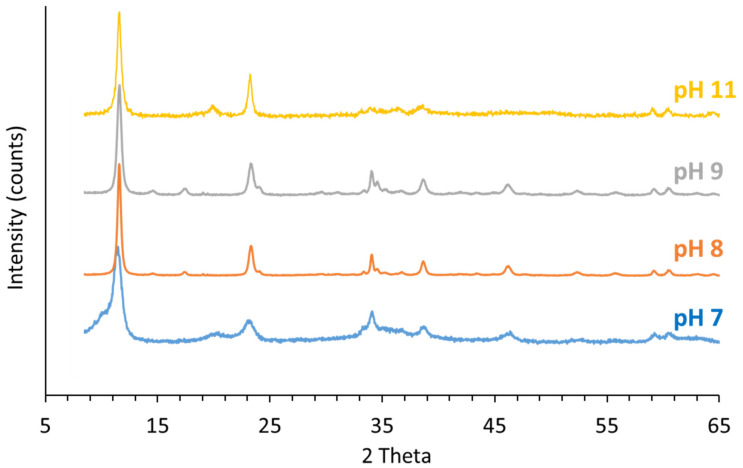
PXRD pattern of the solids collected after aging.

**Figure 5 nanomaterials-12-01570-f005:**
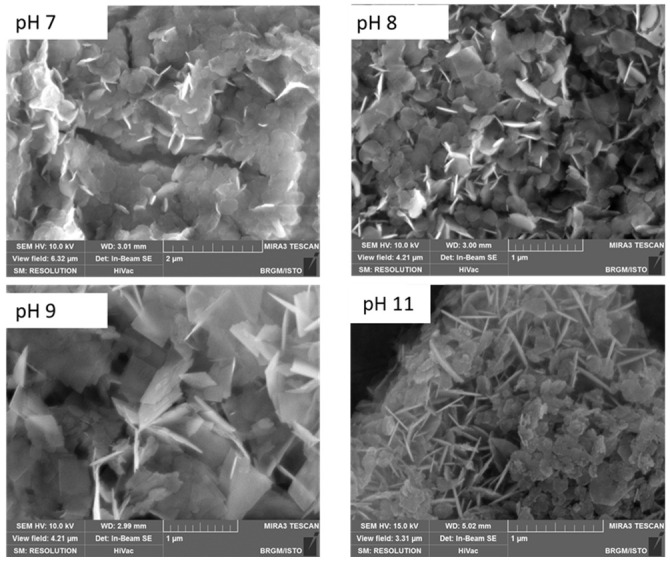
SEM images for Co/Fe LDHs synthesized at pH values 7, 8, 9 and 11.

**Figure 6 nanomaterials-12-01570-f006:**
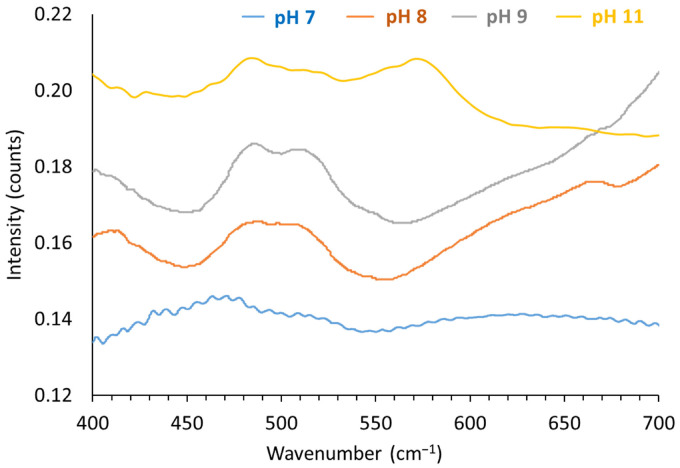
FTIR spectrum of the LDHs synthesized at pH 7, 8, 9 and 11 in the range from 400 to 800 cm^−1^.

**Figure 7 nanomaterials-12-01570-f007:**
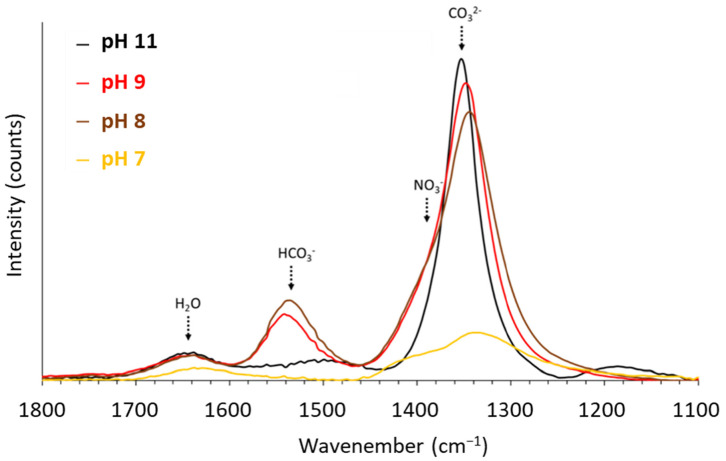
FTIR spectra obtained for the solid collected after the titrations were carried out until reaching pH values 7, 8, 9 and 11.

**Figure 8 nanomaterials-12-01570-f008:**
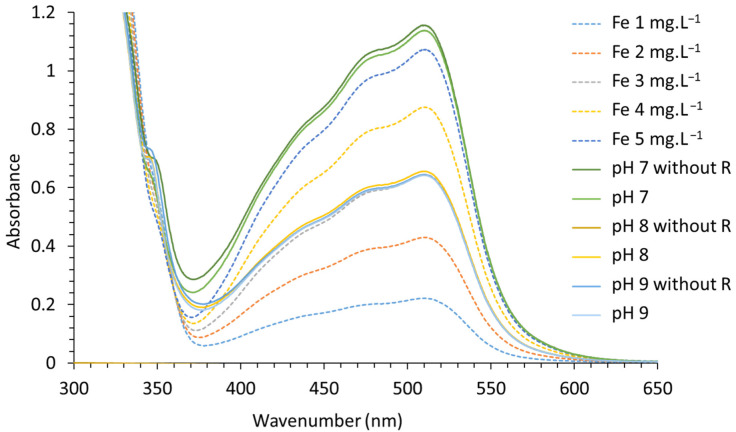
UV–visible absorbance of various concentrations of Fe and synthesized LDHs, with and without the Fe^III^ reducing agent.

**Figure 9 nanomaterials-12-01570-f009:**
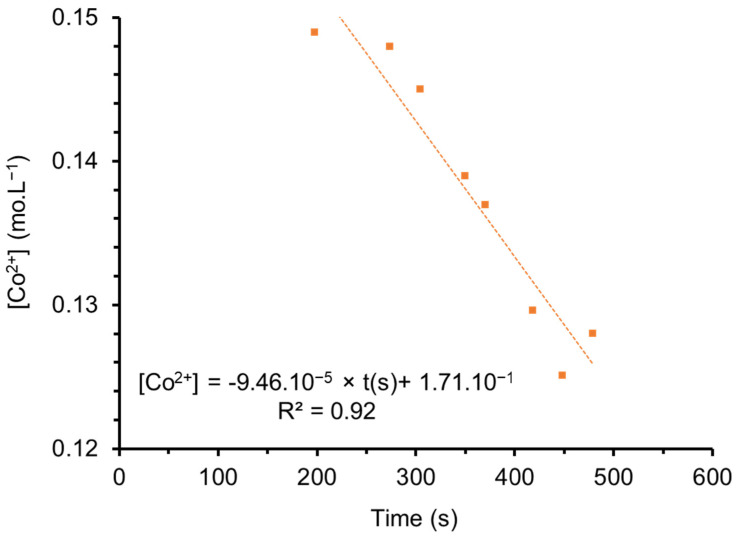
Evolution of the Co concentration versus time from pH 2 to 2.2. Results were provided by AAS and calculated using conductivity measurements.

**Table 1 nanomaterials-12-01570-t001:** XRD data provided by the Rietveld refinement method for the solids obtained from pH 7 to 11.

Characterization Data for Solid Obtained from pH 7 to 11				
pH of synthesis	7	8	9	11
LDH species	2	2	2	2
*d*(003) (Å)	7.64–7.68	7.64–7.64	7.63–7.63	7.63–7.69
*c* parameter (Å)	22.92–23.04	22.92–22.91	22.90–22.90	22.89–23.06
*a* parameter (Å)	3.12–3.09	3.11–3.10	3.08–3.11	3.11–3.09
*t* crystallite size (nm)	12.20–3.20	6.78–3.19	6.00–1.80	10.06–5.29
LDH species relative content (%)	14–86	22–78	16–83	62–38

**Table 2 nanomaterials-12-01570-t002:** XRD data provided by the Rietveld refinement method for the solids obtained from pH 7 to 11 after aging.

Solids Obtained from pH 7 to 11				
pH of synthesis	7	8	9	11
LDH species	3	2	2	3
*d*(003) (Å)	7.62–7.71–8.71	7.61–7.63	7.60–7.63	7.66–7.65–7.72
*c* parameter (Å)	22.86–23.13–26.14	22.84–22.86	22.63–22.90	22.97–22.95–23.15
*a* parameter (Å)	3.12–2.96–3.07	3.12–3.12	3.12–3.12	3.22–3.13–3.06
*t* crystallite size (nm)	16.28–8.14–4.88	52.89–23.12	43.36–19.41	51.57–16.45–4.94
LDH species relative content (%)	34–18–48	84–16	87–13	4–17–79

**Table 3 nanomaterials-12-01570-t003:** LDH particle thickness and length versus pH.

pH	LDH Thickness (nm)	LDH Length (nm)
7	31–36	170–290
8	11–44	180–400
9	27–47	475–510
11	23–42	100–300

## Data Availability

The data presented in this study are available within the article or Appendix A.

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
