# Peer review of "Comprehension of the Route for the Synthesis of Co/Fe LDHs via the Method of Coprecipitation with Varying pH"

_nanomaterials, 2022, doi:10.3390/nano12091570_

Round 1
Reviewer 1 Report
The manuscript by Cherif Morcos and colleagues devoted to the the synthesis
route of Co/Fe LDH by coprecipitation method at variable pH.
The authors have been applied sorption of Co ions by ferrihydrite nanoparticles in th wide pH range (3-11).
To verify the obtained samples the X-Ray diffraction, vibrational and absorption spectroscopies and chemical analyses were used.
In my opinion, the paper is well organized. The experimental section is well described. The results of the synthesis of the LDH
using ferrihydrite nanomaterial as a precursor are adequately presented.
At the same time, some points should be corrected before publication.
First, I am not sure in the neccesity of the chemical equations usage in the Abstract.
Also, I suggest to interpret the LDH abbreviation in the Abstract for the Reader convenience.
2)It is evident from Fig.2 that Co is responsible for the conductivity value.
May the authors explain why the Fe cations do not contribute in the conductivity at low pH value?
3) There are no lattice parameters units in line 263.
4) It is shown the formation of the CoO(OH) in PXRD.
Is it possible to examine Co-O bonding in the FTIR spectra (Fig.5)? In my opinion,
It would amplifies the evidence of the LDH existence.
5) Why a gap of the Co concentration in the 4-20 min interval occurs in Fig.7?
May the authors give the error bars for determining the concentration of the Co?
Author Response
A point-by-point response to the reviewer’s comments was done in the attached word

Reviewer 2 Report
The Authors detailed the Co/Fe-based LDH formation mechanism using a combination of powder X-ray diffraction technique, UV-Vis and FT-IR spectroscopies, plus atomic absorption spectroscopy. The isolation and characterization of different intermediate solid materials following a synthetic protocol based on acid-base titration allowed the Authors proposing a formation mechanism based on precipitation and redox steps. The work is interesting in principle, but I must report some issues that might compromise its quality, and they should be discussed/solved.
My first issue is related to the use of the literature, and I honestly mention the Authors that, in my opinion, the Introduction should be integrated with more references. Although they reported a good number of papers, I was upset not to find a higher number of review articles or book chapters to be cited, considering that the topic is widely investigated and there is a lot of literature to refer to. Reviews present the assessed knowledge, and should be cited with higher priority compared to research articles Only 3 citations over 21 are review articles or book chapters. Moreover, cited articles are not so recent if we look again at the first 21 citations appearing in the introduction (half of them are 15-20 years old, or more), so I ask for an extra effort, pointing at providing the readership an updated list of references.
Some sections, like the Abstract and the Conclusions, should be reshaped, to immediately return the most important information and underline the Authors achievements instead of presenting several minor details (i.e., the abstract should not present all equilibria ruling the formation of Co/Fe LDHs, they are discussed in detail within the manuscript).
Then, there are some issues related to the style in terms of: i) wording and/or sentences organization, and for this reason I left numerous comments along the text giving suggestions of different wording or a different arrangement of the text, to make it clearer and avoiding the risk to have the readers lost. One example is given by PXRD section, where labels from ref. 33 appear, but without the necessary information given to the reader; ii) style, as the text looks very analytical in some passages, but then there are a lot of details missing, which should be reported from an Analytical Chemist, i.e. values uncertainties, standard deviations, etc.
Some concern comes to the FTIR spectra analysis. From a qualitative viewpoint, analysis is correct. But, since there is no external reference peak and no normalization of the spectra, I have a little concern when the Author speak of higher or lower intensities, as it could be a speculation based on the misleading assumption that the samples have been prepared exactly the same way, exactly with the same density of material within the KBr matrix. It could be, it could be not, a reference peak and a normalization work should be evoked before presenting the results
Besides those, there are some minor mistakes or issues to be considered during the cycle of revision, before finalizing the publication to the final step. In the file in attachment, the Authors can find corrections/suggestions evidenced in yellow, and the comment therein reported; I hope the Authors could use them to improve the quality of their research report before moving to the next stage of revisions.

Author Response

(The authors gave the same response as above.)

Reviewer 3 Report
The article Comprehension of the synthesis route of Co/Fe LDH by coprecipitation method at variable pH is devoted to the study of the properties, as well as the stages of formation of Co/Fe structures. Undoubtedly, the results presented by the authors are of high scientific novelty and practical significance, and are also promising for practical research. In general, the presented results of the study can be accepted for publication after the authors provide answers to all the questions raised by the reviewer during the reading of the article.
1. In the abstract, the authors need to more clearly state the purpose and relevance of this work.
2. The authors should explain the choice of components for synthesis, as well as the conditions for obtaining ferrihydrites.
3. Authors should present the results of the phase composition, as well as how they were determined. The numerical ratio of the phase composition should also be presented.
4. Authors should provide size charts of the grains that were used as the objects under study.
5. The authors should explain what is the reason for the change in the size of crystallites, and also how exactly they were determined, is the presented value an average value?
6. Conclusion requires significant revision.
Author Response

(The authors gave the same response as above.)

Round 2
Reviewer 2 Report
I gratefully thank the Authors for the points of discussion they have offered in their reply. I am glad to see that they have positively answered most of my questions/issues. The manuscript is more solid and enjoyable for the readership; the introduction presents a more solid introduction, and all the typo mistakes have been addressed.
For these reasons, I'm in favor of the publication of this newer version. I do not have any further issues to discuss or corrections to point out.
Reviewer 3 Report
The authors answered all the questions, the article can be accepted for publication.